Demand prediction for urban air mobility using deep learning

Ahmed Faheem 1
Memon Muhammad Ali 1
Rajab Khairan 2
Alshahrani Hani 2
Abdalla Mohamed Elmagzoub 3
Rajab Adel adrajab@nu.edu.sa 2
Houe Raymond 4
Shaikh Asadullah 5
1 Department of Information Technology, University of Sindh , Jamshoro , Sindh , Pakistan
2 Department of Computer Science, College of Computer Science and Information Systems, Najran University , Najran , Saudi Arabia
3 Department of Network and Communication Engineering, College of Computer Science and Information Systems, Najran University , Najran , Najran , Saudi Arabia
4 INP-ENIT, Univeristy of Tolouse , Tarbes , France
5 Department of Information Systems College of Computer Science and Information Systems, Najran University , Najran , Saudi Arabia
Kong Xiangjie
Electronic publication date: 2024 Apr 5
Publication date: 2024
Volume: 10
Electronic Location ID: e1946
Received 2023 May 29; Accepted 2024 Feb 28
Copyright: ©2024 Ahmed et al.
Copyright year: 2024
Copyright holder: Ahmed et al.
License: This is an open access article distributed under the terms of the Creative Commons Attribution License, which permits unrestricted use, distribution, reproduction and adaptation in any medium and for any purpose provided that it is properly attributed. For attribution, the original author(s), title, publication source (PeerJ Computer Science) and either DOI or URL of the article must be cited.
License URL: https://creativecommons.org/licenses/by/4.0/

Keywords: Deep learning, Urban air mobility, Prediction, Demand of mobility, Temporal data

Funding: The Deanship of Scientific Research at Najran University under the Research Groups Funding program at Najran University, Kingdom of Saudi Arabia NU/RG/SERC/12/35 The authors received funding from the Deanship of Scientific Research at Najran University for this research through a grant (NU/RG/SERC/12/35) under the Research Groups Funding program at Najran University, Kingdom of Saudi Arabia. The funders had no role in study design, data collection and analysis, decision to publish, or preparation of the manuscript.

==============================
Urban air mobility, also known as UAM, is currently being researched in a variety of metropolitan regions throughout the world as a potential new mode of transport for travelling shorter distances inside a territory. In this article, we investigate whether or not the market can back the necessary financial commitments to deploy UAM. A challenge in defining and addressing a critical phase of such guidance is called a demand forecast problem. To achieve this goal, a deep learning model for forecasting temporal data is proposed. This model is used to find and study the scientific issues involved. A benchmark dataset of 150,000 records was used for this purpose. Our experiments used different state-of-the-art DL models: LSTM, GRU, and Transformer for UAM demand prediction. The transformer showed a high performance with an RMSE of 0.64, allowing decision-makers to analyze the feasibility and viability of their investments.

Introduction

In recent years, we have witnessed the rapid growth of urbanization, representing a significant challenge to the world. An alarming increase in population growth and environmental problems are becoming more acute cases across the globe in “megacities” (United Nations, 2020). The relevant research work was undertaken in the field of UAM and showed that some scientific challenges are yet to be solved, especially in coping with the growing need for on-demand services. To provide valuable advice to policymakers (public authorities) and service providers, we have also concluded that the viability of the market needs to be explored. To this end, this article investigates demand prediction in the context of UAM. In particular, we consider the hypothesis of taxi-based air mobility, using a benchmark dataset for ground taxis provided in the literature. The idea is to propose a predictive model that will be extended to realistic air transport scenarios, considering the transfer learning approach. This article is organized as follows: first, the machine learning pipeline concept and the theoretical background of the main studies on air taxi demand prediction available in the literature is presented. The second section describes the dataset used in this study. The third section presents. The detailed analysis of UAM demand prediction using different state-of-the-art deep learning (DL) approaches, for which respective performances will then be compared. Hence, machine learning methods have limitations over deep learning methods, including reliance on quality and quantity of data, manual feature engineering, and difficulty generalizing to unseen data. They can also be computationally intensive, making them less suitable for handling sequential data. Hyperparameters, such as learning rates and regularization parameters, can significantly impact performance. In real-world datasets, imbalanced data can lead to biased predictions or poor performance on minority classes. Some algorithms may not scale well to large datasets or high-dimensional feature spaces. Additionally, machine-learning models can inherit biases, leading to unfair outcomes in sensitive domains like healthcare or criminal justice. Despite these challenges, machine-learning methods have been successfully applied in various fields and continue to drive innovation in artificial intelligence. After that, the experimental results achieved are presented. The final section describes the principle of transfer learning to propagate the knowledge obtained from the benchmark dataset to any model trained on the new UAM dataset. Finally, the article concludes with a summary of important insights and future research perspectives.

Literature Review

Predicting demand for UAM has proven difficult because the UAM sector lacks historical data. Besides, the other challenge is capital requirements in the near-term market (Zaid, Belmekki & Alouini, 2023). Operating costs are likely to be high, influencing the consumer base. Public perception of air taxis is difficult to determine based on factors, i.e., customer safety, pre-boarding screening, long-distance travel choice, etc. (Goyal et al., 2018) (https://ntrs.nasa.gov/citations/20190001472). Therefore, this work aims to understand the value of demand to the development of UAM and develop a general model to predict more accurate demand for UAM using available taxi data. The following research question is pursued to achieve these objectives: How will demand prediction help UAM business model implementation decision-making for different stakeholders on available taxi data? To answer this research question, we will first discuss various approaches to solving the aforementioned scientific challenges are described in more detail in the following sub-section. Several approaches have been proposed to resolve the aforementioned inherent difficulties and predict customer demand for UAM. Two of these approaches have been discussed in the literature: (i) using qualitative methods to gain the subjective opinions of potential customers and experts, such as focus groups and scientifically structured surveys, and (ii) using quantitive methods to use other mobility provider’s historical data, i.e., taxi service, ride-sharing, ride-hailing service, etc.

Since most of these studies in scientific literature used historical data to predict demand for UAM, very little qualitative research is currently available to predict demand for UAM. While discussing the qualitative approach, Garrow et al. (2018) conducted a survey, and their findings show that rich people will first adopt UAM because it gives more value to time over the cost. One more significant result is that the top three business sectors for attracting clients are inter-city transfers, airport transfers, and frequent travellers. Other surveys are also done by Garrow et al. (2018) to estimate consumers’ willingness to pay for air taxis in densely populated areas of the US. A sampling strategy (Garrow et al., 2019) is used to construct a 100-question survey that considers the impact of several constructs (e.g., lifestyle, personality, perceptual, attitudinal, sociodemographic). These questionnaires were adopted and used a discrete choice model by Boddupalli (2019) to collect the opinions of 2,500 passengers from five major cities of the USA and predict their perception regarding UAM. The study’s main finding is that air taxi demand is very segmented, and the most influential factors were trip time and cost. At the same time, income had no significant influence on planning. The commuting mode can also indicate reasons that impact demand for new means of transportation like UAM. In an urban setting, economic considerations like trip fares and individual income are also the main indicators of mode preference. Ground taxis and urban air mobility (UAM) services have significant price differences due to factors such as infrastructure costs, vehicle costs, operating costs, regulatory compliance, market dynamics, and technology development. Ground taxis use existing road infrastructure, while UAM services require significant investments in infrastructure and eVTOL aircraft. UAM services may have higher operating costs, including electricity, maintenance, pilot training, and insurance. Regulatory compliance could increase operational costs. Market dynamics may limit demand for UAM services, while ground taxis may benefit from a larger customer base and economies of scale. Technology development is still in its early stages, but economies of scale could reduce costs over time. Despite these challenges, ground taxis offer faster and more efficient urban mobility. Interestingly, the preferences of higher-income people are mixed, with some preferring private vehicles and some preferring on-demand taxis (Sivasubramaniyam, Charlton & Sargisson, 2020). Some other impacts are listed in the scientific studies, i.e., parking space and crowding (Ye & Titheridge, 2017). In addition to economic factors, age and gender affected transportation modes. According to the study of Chee & Fernandez (2013), males use more public transport than females. As an age example, Tyrinopoulos & Antoniou (2013) discovered that commuters aged 35 to 44 had a higher preference for driving their vehicle. This qualitative research will determine infrastructure requirements, such as the location of these vertiports, the number of vertiports required, pricing strategies for fleet procurement, and the number of air taxis to be procured or manufactured during the UAM implementation process, among other things (Schweiger & Preis, 2022; Yunus et al., 2023) The fact that these high-risk actions are expected to have a negative impact on revenue and efficiency means that they will have an impact on UAM service providers’ medium to long-term strategies (Boo, Lee & Song, 2023).

Regarding sustainable development goals (SDGs) discussed in the literature, Mudumba et al. (2021) presented an emissions computation framework for predicting CO2 emissions for any UAM trip. The framework calculates trip emissions using UAM for part or all of the journey and compares them to trips made by electric cars. The results show that a region’s grid emission index significantly impacts how much CO2 is released into the air when an electric vehicle is used in the Chicago and Dallas metropolitan areas. As of the region’s low grid emission index, trips involving electric vehicles such as non-autonomous electric vehicles (BEV) and UAM is more environmentally friendly than gasoline automobiles in the Chicago region. Because of the high grid emission index in the Dallas region, the benefits of using UAM are not as noticeable as the benefits of driving a gasoline automobile in this region. Since UAM services are in the experimental stage, it may be possible to use analytical methods used to predict demand once these services are operational. Few studies use traditional transportation mode data to predict future demand. The machine learning method is applied to time-series data by Moreira-Matias et al. (2013), Tahseen et al. (2022) and Jiang et al. (2019) to predict the count of short-distance passenger requests (5–60 min) for online car-hailing services based on the LS-SVM model. However, since they do not use a trend model, it is difficult to improve it further. Two deep learning architectures (Rajendran & Zack, 2019) are used to combine text knowledge with time-series data from multiple sources and use word embeddings, convolutional layers, and attention mechanisms to increase prediction accuracy. The first to use a gravity model to forecast long-term consumer demand for UAM was used (Becker et al., 2018). They found 26 possible high-demand markets for UAM implementation worldwide based on eight socioeconomic variables. This research can be combined with other research, such as Uber Elevate, to evaluate additional criteria, such as potential customers or cost, to obtain more accurate results regarding potential markets. A multi-task deep learning (MTDL) model (Luo et al., 2021) is proposed to predict the demand for a short-distance taxi at different zones, which can predict the volume of demand in numerous traffic zones in real time. This model outperforms other prediction models by comparing the following evaluation metrics, i.e., MAPE, RMSE, R2, and MAE, due to its ability to dig the spatiotemporal characteristics in a large data sample. This research is only useful while predicting short-distance taxi demand; it does not apply to long-distance real-time taxi demand. To predict short and long-distance demand for air taxis, Rajendran & Zack (2019) proposed a generalized model based on standard taxi trip data. The first step is to estimate demand for air taxi services using two methodologies by selecting a small number of frequent taxi customers who want to use this service. The second method is to classify potential sites for locating air taxi infrastructure using a clustering approach combined with a multimodal transportation-dependent warm start technique. Results show an effect of commuters’ willingness to travel rates, demand contentment rates, and time-cost tradeoffs. The percentage of saved time and the rate of “willingness to fly” had no substantial effect on location decisions or the number of locations, suggesting a thorough market study is needed to assess on-demand travel. Any service provider interested in entering the air taxi business might utilize the findings of this study as a decision-making tool. One of the study’s limitations is the unavailability of data from other demand sources, i.e., Uber, Lyft, and other transport modes, e.g., buses, trains, and other transit services (Metro, tram, etc.). A discrete event simulation model was proposed by Rajendran & Shulman (2020), leveraging the six sigmas Define, Measure, Analyze, Design, and Verify (DMADV) methodology. The results show that the number of air taxis required to meet the demand for urban air mobility in New York City and all indicators have a linear impact on vehicle usage. Still, other indicators, particularly the seating capacity, are stable. The main failing of the study is the lack of data, and the latent demand that would emerge from the mature air taxi system was not considered. Machine learning algorithms (Rajendran, Srinivas & Grimshaw, 2021) are used to predict the demand for air taxis at different times of the day in different traffic zones of New York City. Demand is predicted using machine learning algorithms such as logistic regression, artificial neural networks, random forests, and gradient boosting. The experiments show that the gradient boosting algorithm provides better prediction accuracy performance. This study delimits MLAs to four well-known algorithms available in the literature. Based on these limitations, we will implement a deep neural network model in this research work to improve prediction performance.

Methodology

This section describes the methodology of demand prediction. Before going into details on UAM demand prediction, it is important to understand the machine learning pipeline concept, which includes many steps for predicting demand.

Machine learning pipeline

A machine learning pipeline is a way of formalizing and organizing the processes required to deploy a machine learning model. Machine learning pipelines are sequential steps that handle everything from data extraction and preprocessing to model training and deployment. The one presented in this section is concerned with supervised learning, using tabular data, as illustrated in Table 1. In this table, Xj (j = 1,2,…) represents a given input feature, and y represents the output (the variable to predict), a continuous variable in our case study.

Table 1 General structure of tabular data in supervised learning.

Sample	X1	X2	…	Y	
1					
2					

The following steps have to be followed to accomplish a demand prediction task. It starts (as shown in Fig. 1) with step 1, data acquisition that includes data collection from different sources, e.g., sensors, physical measurements, and analysis (physical, chemical, etc.). Since urban mobility uses and generates a huge amount of data, we can consider big data, real-time data, and velocity of data as critical aspects while storing this huge data.

Figure 1 Machine learning pipeline.

Then, some data fusion techniques must be performed to build a dataset for the next step, 2, data preprocessing. First, we clean the data to detect and fix incorrect or missing values in the data pre-processing step. For example, we may discover many data anomalies with zero, negative, or illogical values. We can also do feature engineering in this step, which means we can perform some calculations to create a new dataset feature. This step allows putting the data in a format compatible with the machine learning algorithms used in step 3. The third step of the pipeline consists of building or using an appropriate machine learning model, e.g., (support vector machine (SVM), k-nearest neighbor (KNN), Random Forest, etc.), which depends on many parameters like type of data (image, text, voice, time-series, etc.) In step 4, these models are trained and tested by setting different tunning hyperparameters to achieve an efficient performance. For example, in this study, we will use the following hyperparameters:

Embedding size: An embedding size is a relatively low-dimensional space into which you can translate high-dimensional vectors. Embeddings make it easier to do machine learning on large inputs like sparse vectors representing words.

Hidden layers: Hidden layers are the layers between the input layer and output layer.

Batch size: Mini batch size is the number of sub-samples given to the network after which parameter update happens.

Learning rate: The learning rate defines how quickly a network updates its parameters.

Epochs: The number of epochs is the number of times the whole training data is shown to the network while training. In step 5, we used different evaluation metrics to evaluate the performance of implemented models. When evaluating machine learning models, choosing the right metric is also critical. It aims to estimate the generalization accuracy of the future (unseen/out-of-sample) data. For example, for the classification problem, we have matrics such as confusion matrix, precision, recall, F1 score, Log loss, and for the regression problem, we have mean absolute error (MAE), mean square error (MSE), and root mean square error (RMSE). This study uses RMSE as an evaluation metric to evaluate performance. The RMSE is the most commonly used metric for regression tasks (see Eq. 1). This is calculated by taking the square root of the average squared distance between the actual and predicted scores. (1) RMSE1n∑i=1nyi−y ˆi2.

Here, yi denotes the true score for the ith data point, and yi denotes the predicted value.

Ground taxis face longer routing and travel times due to factors such as traffic congestion, route restrictions, stop-and-go nature, terrain and geography, and limited speed. These factors can slow down travel times, especially in densely populated urban areas with congested roadways. Ground taxis also need to navigate around one-way streets, road closures, and detours, adding distance and time. They also need to stop at traffic lights, intersections, or pick-up/drop-off points, extending travel times. Geographic obstacles like rivers, mountains, or densely populated areas can limit route options and require longer detours. Ground taxis are also subject to speed limits, limiting their ability to travel quickly. Air taxis offer time savings and efficiency in urban mobility. The following block diagram shows the basic operation of UAM is illustrated in Fig. 2.

Figure 2 UAM operations (Vaswani et al., 2017).

Dataset of the case study

We analyze our proposed approach using an expected potential demand because air taxi services are currently in the experimental stage and have yet to be launched in megacities. Therefore, we use a percentage of present normal taxi demand using the land taxi-based benchmark dataset to predict future demand for air taxi services. The Uber Elevate white paper information was used to create this demand segment (Holden & Goel, 2016), whereby the company discovered after conducting extensive market research. We examined 150,000 of the New York City (NYC) Taxi and Limousine Commission data records available online for our investigation. UAM is still in the experimental stage; hence, we could not find the exact dataset for UAM to apply our techniques. We consider that NYC taxi data is close in resemblance to the UAM that will be used in the future. One potential limitation is that many taxi trips are within Manhattan between uptown, midtown, and downtown. Relatively speaking, these are short distances with high access to subways. It seems less likely that UAM would serve this type of trip vs. longer distances/travel times from other boroughs to Manhattan. In the future, when this dataset or any other provides long-distance trips, just not apply our model on the dataset to get the results. Another limitation is the NYC dataset used is not of actual UAM but it is the closest one behavior like ground taxi.

Dataset description

This section presents a detailed description of the datasets used in our experimental study. We used the New York City Taxi and Limousine Commission Trip Record Data dataset. This dataset contains records for each taxi journey, comprising pick-up and drop-off, date and time, pick-up and drop-off locations, trip lengths, categorized prices, rate kinds, payment methods, and driver-reported passenger counts (see Table 2). It should be noted that our case study is based on supervised learning. To that end, we should define, among the available variables, the one representing the output (y). In this study, the concerning output is the “demand” that we need to predict. But, in the raw data, no information regarding demand is provided. Therefore, we should define one, which is explained in the next section.

Table 2 Dataset description.

Feature variable	Value examples	
Id: A unique identifier for each trip	1,2, …	
Vendor id: A code indicating the provider associated with the trip record	1, 2	
Pickup date-time: Date and time when the meter was engaged	12/1/2020 0:07	
Dropoff date-time: Date and time when the meter was disengaged	12/1/2020 0:18	
Store and forward flag: This indicates whether the trip record was held in vehicle memory	
before sending it to the vendor. Y=store and forward; N=not a store and forward trip	N or Y	
PU Location ID: The pick-up location id	138	
DO Location ID: The drop-off location id	263	
Payment type: Card, Cash, etc.	1, 2, 3	
1 = Bank card;	
2 = Cash;	
3 = No charge;	
4 = Dispute;	
5 = Unknown;	
6 = Trip canceled.	
Fare amount: Amount of charges in dollars	21.5	
MTA tax: A tax of 0.50 USD MTA has been triggered automatically according to the tariff	
displayed on the meter.	0.5 or -0.5	
Tip amount: Amount to pay in terms of TIP	2.5	
Tolls amount: Toll tax amount depends on the distance	6.12 or 0	
Improvement surcharge: Constant surcharge amount	0.3	
Total amount: fare amount+extra+mta tax+tip amount+toll	
amount+improvement surcharge	33.92	
Passenger count: The number of passengers in the vehicle (driver entered value)	2	
Trip distance: Duration of the trip in minutes	7.6	

Demand calculation

In literature, Rajendran, Srinivas & Grimshaw (2021) uses the density of passengers requesting air taxi rides to travel to a specific location to calculate demand level. Therefore, we use the same columns to calculate the demand ratio based on passenger count and distance (see Eq. 2). The high ratio indicates high demand, and the low ratio indicates low demand. (2) Ratio=PassengerCount∗Distance.

In the next section, we prepare a detailed analysis of UAM demand prediction using different State-of-the-art deep learning (DL) approaches.

Deep Learning-based implemented models

Deep learning is a subfield of machine learning that deals with ”artificial neural networks,” which are algorithms inspired by the structure and function of the brain. In other words, it mirrors how human brains work. Deep learning algorithms are structured similarly to the nervous system, with each neuron connecting and passing information. Deep learning models work in layers, and a typical model atleast has three layers. Each layer accepts the previous information and passes it on to the next one. Deep learning models tend to perform well with the amount of data, whereas old machine learning models stop improving after saturation. In this section, we use three state-of-the-art deep learning-based approaches, including long-short term memory (LSTM) (Hochreiter & Schmidhuber, 1997), gated recurrent unit (GRU) (Jozefowicz, Zaremba & Sutskever, 2015), and Transformer (Vaswani et al., 2017) for the UAM demand prediction task. We will also consider these as baseline classifiers to compare and use these networks as input (transfer-learning) to a new model.

Long short-term memory (LSTM)

We implement the long short-term memory (LSTM) network, an improved variant of recurrent neural networks, to remember previously-stored information. This method was used to solve the RNN gradient problem. The LSTM method is well-suited for identifying, processing, and predicting time series with unknown time lags. We trained LSTM via backpropagation and reduced the network loss. An LSTM network has three gates see (Fig. 3).

Figure 3 Long short-term memory (LSTM) gates (Phi, 2018).

Input gate: Discover which value to change memory from the input. The sigmoid function determines what values [0,1] should be moved. The tanh function, on the other hand, assigns weight to the values passed to assess their importance on a scale of [−1 to 1]. (3) it=σWi∗ht−1,xt+bi

(4) C ~t= tanhWc∗ht−1,xt+bc

Forget gate: Discover what blocks discarding information. The sigmoid function is defined as it examines the previous state and content inputs and returns a number ranging from 0 (omit this) to 1 (keep this) for each of the inputs C(t−1) cell state number. (5) ft=σWf∗ht−1,xt+bf

Output gate: Block input and memory to evaluate output. The sigmoid function defines 0,1 values. The tanh function gives weighting to pass values to determine their level of importance from −1 to 1 and multiplied with sigmoid output. (6) Ot=σWo∗ht−1,xt+bo

(7) ht=Ot∗ tanhCt.

Gated recurrent unit (GRU)

We used the GRU network, the enhanced version of the regular recurrent neural network (Cho et al., 2015). GRU employs gate updates and resets to solve the vanishing gradient problem of a standard RNN. These two vectors determine the information that should be transferred to the output. They are unique in that they can be trained to remember past information without being washed away over time or information unrelated to prediction being eliminated. Figure 4 demonstrates detailed gated recurrent unit (GRU) architecture.

Figure 4 Gated recurrent unit (Phi, 2018).

Update gate: For time step t, the formula for calculating the update gate zt is: (8) zt=σWzxt+Uzht−1.

Next, xt is connected to the network unit, it is multiplied by its weight W((z)). Similarly for h(t−1) stores data for the preceding t-1 units then multiplied by its weight U((z)). All outcomes are entered, and the number is squashed between 0 and 1, applying a sigmoid activation feature. Figure 5 follows the schema as mentioned earlier.

Figure 5 Illustrate the schema of the update gate (Phi, 2018).

The model can use the update gate to determine how much previous knowledge (from earlier steps in time) should be transferred into the future. This is extremely useful because the model will decide to copy all previous data and eliminate the possibility of gradient fading.

Reset gate: Basically, the model uses this gate to determine how much past knowledge to forget. To measure, we applied: (9) rt=σWrxt+Urht−1

This formula is the same for update gates. The difference is in weight and gate use. Figure 6 below indicates where the gate is: As previously, we plug in h(t−1) blueline and xt purple line, multiply with their respective weights, add the outcomes, and use the sigmoid function.

Current memory content: The effect of the gates on the final product is demonstrated in this example. Then, we start using the reset gates. The reset gate stores past data and adds new memory content. The following is how it was calculated: (10) ht′= tanhWxt+rt⊙Uht−1

Figure 6 Illustrate the schema of the reset gate (Phi, 2018).

Step 1. The input xt multiply by a weight W and h(t−1) by a weight U.

Step 2. Hadamard (element-wise) calculates among the reset gate rt and Uh(t−1). It resolves to decide what measures should be removed from the previous ones. Assume we are dealing with a sentiment analysis issue to determine someone’s feelings for a book review. After a few paragraphs, the text begins with “This is a fantasy novel that explains...” and ends with “I didn’t like the book because I think it captures so many specifics.” We only need the last section of the review to determine the book’s overall satisfaction level. A neural network gets closer to the conclusion of the text; this will be learning to allocate rt vector near to 0, trying to erase previous sentences and concentrating only on the ending sentences.

Step 3. Steps 1 and 2 should be added together.

Step 4. Use the tanh nonlinear activation function.

Figure 7 Illustrate the current memory content (Phi, 2018).

Figure 7 illustrates the steps. We ensure an element-wise multiplication of h(t−1) blueline, and rt orange line and then add the result pink line by the input xt purple line. Lastly, tanh is used to produce ht′ bright green line.

Final memory at current time step: Lastly, the network, represented by the, ht vector is measured. This vector contains information about the current unit and transfers it to the network. Therefore, the upgrade gate is required. Defines what should be collected from present memory substance ht′ and what should be collected from prior steps h(t−1). This is how it’s done: (11) ht=zt⊙ht−1+1−zt⊙ht′.

To update the gate, element-wise multiplication should be used zt and h(t−1). Use element-wise multiplication to 1 − zt and ht′. Add the outcomes from steps 1 and 2 The most critical information is now at the start of the document. It will learn the vector environment, zt near to 1 and hold the most previous details. After that, zt approximately 1 at this time stage, 1 − zt approximately 0 would disregard a significant percentage of the remaining content (in this scenario, the last section of the analysis describing the book plan), insignificant to our prediction. Figure 8 highlights the above equation: As a result of this, zt the green line was used to compute 1 − zt and combined with ht′, the bright green line produces a result as a dark red line. zt is used with h(t−1) blueline for element-wise multiplication. Lastly, ht, the blue line results from the summation of the outputs equivalent to the bright and dark red lines. Hence, we can see how GRUs can store and filter data through their gates. The vanishing gradient problem is no longer an issue because the model does not wash out the new input every time it is run. In its place, it saves the relevant data and forwards it to the next network step. They can perform exceptionally well in complex situations if they have received the proper training and preparation.

Figure 8 Illustrate the final memory content (Phi, 2018).

Transformer

The “Attention Is All You Need” article introduces a new architecture known as “Transformer” (Vaswani et al., 2017). Similar to the LSTM, Transformer is a two-part architecture for transforming sequences used in machine learning (encoder and decoder). Since it does not imply recurrent networks, it differs from previous sequence-to-sequence models (GRU, LSTM, etc.). Using recurrent networks to capture sequence dependencies is one of the most effective methods available as shown in Fig. 9. Architecture is shown that on the left is the encoder, and on the right is the decoder. In the diagram, Nx shows the encoder and decoder as stackable modules. Each module is comprised of feed-forward and multi-head attention layers. Because we cannot use strings directly, we must first embed the inputs and outputs in an n-dimensional space. The model’s positional encoding of words is a minor but crucial component. We must assign a relative position to each word or part because the order of the elements in a sequence is important. Each word contains these positions (n-dimensional vector) (See Fig. 10). Let us start with the attention mechanism’s description. It is simple and can be described by the equation: (12) AttentionQ,K,V=σQKidkV.

Figure 9 Transformer model architecture (Vaswani et al., 2017).

Figure 10 Scaled dot-product attention and multi-head attention (Vaswani et al., 2017).

Q is a matrix containing the query (vector representation of one word in the sequence), K contains all the keys (vector representations of all words in the sequence), and V contains all the values. V and Q are the same word sequences for multi-head attention modules in encoders and decoders. V is different from Q because it is used in the attention module, which considers encoder and decoder sequences. So, we multiply and add the values in V together with certain attention weights specified as follows: (13) a= softmaxQKTdk.

The weights are based on how each word (Q) influences the others (represented by K). Weights between 0 and 1 are also applied to the SoftMax function. Those weights are used for all words in V. When compared to Q, the encoder and decoder vectors are the same but different from the module with both encoder and decoder inputs. It is possible to parallelize this attention mechanism into multiple mechanisms, as illustrated in the right image. The attention mechanism is recurrent, having the types Q, K, and V linear projections. As a result, the model may learn from dissimilar Q, K, and V representations. These linear representations multiply the variables Q, K, and V by learned weight matrices W. Q, K, and V of the attention modules differ dependent on whether they are located in the encoder, decoder, or somewhere between these two locations. We need to focus on also the entire encoder input sequence or a subset of it for this analysis. A multi-head attention module is used to ensure that an encoder input sequence and the decoder input sequence are considered up to a certain point in the decoding process. A pointwise feed-forward layer follows the encoder and decoder’s multi-attention heads. Using the same parameters for each position in the sequence, this small feed-forward network performs a linear transformation on each element in the given sequence.

Experimental Results and Discussion

In this section, we first share the execution environment configuration before presenting the results of the implemented models, namely LSTM, GRU, and transformer discussing some key implementation insights.

Execution environment

We set up the following execution environment to implement the aforementioned deep learning models:

• Python 3 was used as a programming language.

• System characteristics used an Intel Core i5 processor at 2.67 GHz and 8GB of memory.

• Libraries used for implementation are TensorFlow (Dean & Monga, 2015), Scikit-learn (Vaswani et al., 2017), and Keras (Cho et al., 2015).

Results and discussion

In this section, the results of each step of the machine-learning pipeline discussed earlier in the methodology section are presented in detail.

Step 1. Data acquisition: Data acquisition is the first step in the machine learning pipeline. Since air taxi services are currently in the testing phase, we do not have the actual data for demand prediction. As a result, we predict future demand for air taxi services using a percentage of current normal taxi demand from the land taxi-based benchmark dataset. Therefore, we use the New York City Taxi, and Limousine Commission Trip Record Dataset in this study discussed earlier in the ‘Dataset description’ section, with a detailed description of each field of the dataset. This dataset contains records for each taxi journey, comprising pick-up and drop-off, date and time, pick-up and drop-off locations, trip lengths, categorized prices, rate kinds, payment methods, and driver-reported passenger counts.

Step 2. Data preprocessing:

The data preprocessing is the second step and critical step in the machine learning pipeline. We discussed the case study dataset and the description of each dataset column in the ‘Literature Review’ section. Since this dataset is a raw dataset, we found many data anomalies that need to be fixed in this step, e.g., a few columns with zero, negative, and illogical values. We also eliminate some columns in this data preprocessing steps that are no longer useful for demand prediction (rate_code id, congestion surcharge, etc.)

Step 3. Models implemented: In this step, we implement three deep learning-based models (LSTM, GRU, and Transformer) that were tested in our experiment, and the results were as follows:

LSTM trained on the training dataset and tested on the test dataset.

GRU trained on the training dataset and tested on the test dataset.

The transformer was trained on the training dataset and tested on the test dataset.

Step 4. Models training: In this step, we trained and tested the models mentioned earlier by using different hyperparameters to tune the performance of the methods discussed above in step 3.

Tables 3, 4 and 5 shows the hyperparameters for the LSTM, GRU, and transformer network.

Table 3 Hyperparameters for LSTM network.

Hyperparameters	Values	
Embedding size	128	
LSTM layers	04	
Batch size	64	
Learning rate	0.0001	
Epochs	100	

Table 4 Hyperparameters for GRU network.

Hyperparameters	Values	
Embedding size	128	
LSTM layers	03	
Batch size	64	
Learning rate	0.0001	
Epochs	100	

Table 5 Hyperparameters for transformer network.

Hyperparameters	Values	
Hidden_Size	16	
Attention_Head_Size	1	
Dropout	0.1	
Hidden_Continues_Size	8	
Loss	QuantileLoss	
Learning rate	0.03	
Output_Size	7	
Epochs	100	

Step 5. Models evaluation

As previously discussed in the ‘Methodology’ section, we used various evaluation metrics to assess the performance of the implemented models, including mean absolute error (MAE), mean square error (MSE), and root mean square error (RMSE) (RMSE). We used RMSE as an evaluation metric to evaluate performance in this experiment. For regression tasks, the RMSE is the most commonly used metric. The square root of the average squared distance between the actual and predicted scores is used to calculate this. As a result, we achieved a root-mean-square error of 2.06 for the LSTM network and 2.11 for the GRU network and 0.64 for the Transformer network. The experimental results show that the transformer outperformed on GRU and LSTM networks. The detailed results are summarized in Table 6. Our empirical study presented the evidence to support the demand prediction with relatively high predictive performance. Our experimental result shows that a Transformer was applied to analyse urban mobility demand and highly predictive performance (Abbasi et al., 2020; Abassi et al., 2023). Since the Transformer method performs better than GRU and LSTM, we will analyze some key insights in detail in the next section.

Table 6 Performance comparison between LSTM, GRU, and Transformer.

Method	RMSE	
LSTM	52.06	
GRU	2.11	
Transformer	0.64	

More results of the best model

As previously demonstrated, the transformer model outperformed LSTM and GRU. More transformer results, specifically attention loss, learning curve, and variable importance, are examined in this section.

Attention loss. Regarding attention loss for training and testing steps, we have seen in Figs. 11 and 12 that values are overlapping. It shows that the loss is relatively minimal between observed and predicted values.

Figure 11 Shows the testing loss.

Figure 12 Shows the training loss.

Learning curve. We ran multiple experiments with different values to find the optimal learning rate and found that a 0.035 learning rate was the best (see Fig. 13).

Figure 13 Shows the learning rates.

Variable importance. Since we have many variables in our dataset and they do not influence demand equally, that is why we use variable importance to identify the most important variables. Figure 14 depicts important variables that are more useful and influential for forecasting demand, such as trip distance, passenger count, and pickup and drop-off locations. Fig. 13). It is interesting for decision-makers who want to deploy or implement UAM. Understanding and achieving optimal values for these variables is critical when investing. For example, a high passenger count indicates a high demand for long enough distances. Pick-up and drop-off locations are also important for decision-makers when deciding on new UAM infrastructure deployment, such as vertiports, etc. These significant findings indicate the possibility of utilizing an approach such as transfer learning to propagate knowledge from previously trained models to a new UAM dataset, as outlined in the following section.

Figure 14 Shows the importance of the decoder features.

Knowledge Transfer from land to air mobility domain

Transfer learning is the process of transferring knowledge using parameters of a neural network trained on a single dataset and to a new domain with a different (Ozbulak, Aytar & Ekenel, 2016). An interesting phenomenon is observed in several deep neural networks trained on numerical data. They learn general features on the first layers of the network, which do not appear restricted to a single dataset but apply to a wide range of datasets. Transfer learning should allow us to apply knowledge from previously learned tasks to newer, related ones. If we have more data for task T1, we can use it to learn and apply this knowledge (features, weights) to task T2 (which has significantly less data). The main motivation, particularly in the context of deep learning, is that most models that solve complex problems require a large amount of data, and obtaining large amounts of labelled data for supervised models can be extremely difficult, given the time and effort needed to label data points. Conventional machine learning and deep learning algorithms have traditionally been designed to work in isolation, which is still the case today. These algorithms are developed over time to perform specific tasks. When the source dataset shifts to reflect the new dataset, the models must be rebuilt from scratch. Regarding transfer learning, the concept refers to overcoming the isolated learning paradigm and applying knowledge gained from one task to solve related tasks. Transfer learning can be a powerful tool for training a large target network without overfitting it when the target dataset is significantly smaller than the base dataset. In light of the above result, we can use the transfer knowledge from the above well-trained architecture, such as the transformer, to the new model and train with the UAM dataset. This approach will boost the learning curve of the new model for UAM demand prediction and reduce the under and overfitting problems. Figure 15 shows the high-level architecture of the new model with a transfer learning approach.

Figure 15 UAM demand prediction using transfer learning.

Conclusions and Future Work

This article investigated the demand prediction of a taxi UAM use case to provide policymakers and service providers with in-depth knowledge of the return on investment required for UAM deployment. A benchmark dataset of 150,000 records was used for this purpose. Our experiments used different state-of-the-art DL models: LSTM, GRU, and Transformer for UAM demand prediction. The transformer showed a high performance with an RMSE of 0.64, allowing decision-makers to analyze the feasibility and viability of their investments. This valuable result demonstrates the potential for leveraging an approach such as transfer learning by propagating knowledge from previously trained models to a new UAM dataset. We believe this will improve prediction results compared to other basic networks trained on raw data. In the future, the purpose of this research will be to evaluate the practicability of deployment. We will be able to suggest a data-driven framework for the operation of UAM, highlighting the interoperability difficulties present in its primary components. It is considered, that this work is limited by the NYC dataset used, which is not of the actual UAM dataset but it is the closest one behaviour like a ground taxi. Additionally, proposed techniques are executed using third-party high-performance GPUs, which could be improved further with more training, testing and even with other deep learning techniques. Based on the findings, UAM services should be integrated with existing public transportation systems, and safety and risk management should be prioritized. Market research should assess the demand for UAM services and its cost-effectiveness. Public policy and governance should address legal, ethical, and social issues related to UAM. A multidisciplinary approach involving government agencies, industry partners, academic institutions, and community stakeholders is crucial for achieving UAM’s full potential in urban areas. Based on the findings, UAM services should be integrated with existing public transportation systems, and safety and risk management should be prioritized. Market research should assess the demand for UAM services and its cost-effectiveness. Public policy and governance should address legal, ethical, and social issues related to UAM. A multidisciplinary approach involving government agencies, industry partners, academic institutions, and community stakeholders is crucial for achieving UAM’s full potential in urban areas.

Supplemental Information

Supplemental Information 1 Trip Data: 2020-2012

Supplemental Information 2 Temporal Fusion Transformer Data

Supplemental Information 3 Taxi Monthly PreProcess

Additional Information and Declarations

Competing Interests

Author Contributions

Data Availability

The authors declare there are no competing interests.

Faheem Ahmed conceived and designed the experiments, analyzed the data, prepared figures and/or tables, and approved the final draft.

Muhammad Ali Memon conceived and designed the experiments, analyzed the data, prepared figures and/or tables, and approved the final draft.

Khairan Rajab conceived and designed the experiments, analyzed the data, prepared figures and/or tables, and approved the final draft.

Hani Alshahrani conceived and designed the experiments, performed the experiments, analyzed the data, prepared figures and/or tables, and approved the final draft.

Mohamed Elmagzoub Abdalla conceived and designed the experiments, performed the experiments, performed the computation work, prepared figures and/or tables, and approved the final draft.

Adel Rajab performed the experiments, performed the computation work, authored or reviewed drafts of the article, and approved the final draft.

Raymond Houe performed the experiments, performed the computation work, authored or reviewed drafts of the article, and approved the final draft.

Asadullah Shaikh performed the experiments, performed the computation work, authored or reviewed drafts of the article, and approved the final draft.

The following information was supplied regarding data availability:

Third Party Data is available at TLC Trip Record Data: https://www.nyc.gov/site/tlc/about/tlc-trip-record-data.page.

The 3D mobility code and dataset is available at GitHub and Zenodo:

- https://github.com/mamemon/3dmobility.git.

- mamemon. (2023). mamemon/3dmobility: Initial Release (0.1). Zenodo. https://doi.org/10.5281/zenodo.10408631.

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
