# Peer review of "Demand prediction for urban air mobility using deep learning"

_PeerJ Computer Science, doi:10.7717/peerj-cs.1946_

## Round 0.1 · original submission · Major Revisions

The reviewers agree the work is interesting and has value. However, they also pointed out that the article needs improvement in writing, experiment design, etc. Please revise it according to the comments.

**Language Note:** PeerJ staff have identified that the English language needs to be improved. When you prepare your next revision, please either (i) have a colleague who is proficient in English and familiar with the subject matter review your manuscript, or (ii) contact a professional editing service to review your manuscript. PeerJ can provide language editing services - you can contact us at copyediting@peerj.com for pricing (be sure to provide your manuscript number and title). – PeerJ Staff

Reviewer 1 ·

Basic reporting

Thank you for the opportunity to review this paper on demand prediction for urban air mobility (UAM). Overall, this is a very interesting paper. A few questions and comments for the authors:

- The use of the term "ridesharing" is a bit controversial, as its often misused by the popular press. The authors may consider removing or replacing with "ridehailing" as appropriate.
- Should the title and abstract reflect that this is a case study using NYC data?

Study Limitations:
- What are the limitations of using NYC TLC data?
- One potential limitation is that a lot of taxi trips are within Manhattan between uptown, midtown, and downtown. Relatively speaking, these are short distances with high access to subways. It seems less likely that UAM would serve this type of trip vs. longer distances/travel times from other boroughs to Manhattan.
- It would be helpful to discuss the limitations in greater detail and how these were overcome and/or recommendations for additional study, as appropriate.

References:
A few minor comments regarding Goyal et al. I was not able to easily find this study? Perhaps the reference can include the publisher and/or weblink, as appropriate based on the citation format.
https://ntrs.nasa.gov/citations/20190001472

A few other resources that may be helpful for the authors:
https://ieeexplore.ieee.org/document/9447255
https://ntrs.nasa.gov/citations/20190026762
https://www.mdpi.com/2071-1050/13/13/7421
https://www.airbus.com/sites/g/files/jlcbta136/files/2022-07/Airbus-UTM-public-perception-study%20-urban-air-mobility.pdf
https://escholarship.org/uc/item/7p69d2bg

Experimental design

No comment

Validity of the findings

No comment

Cite this review as

Reviewer 2 ·

Basic reporting

The authors have proposed "Demand prediction for urban air mobility using deep
learning approach with id (#85919)".
The authors are required to explain the following point wise explanation.
1) With respect to the existing literature how your proposed models differs. Novelty of the work is missing. It should be strongly highlighted. As all the models exist in literature how the authors would justify the significance of the proposed model. If any fine tuning has been done it need to be specified.
2) Make some relevant and state of the art comparisons with the proposed method.
3) The findings should be validated with different datasets.
4) the Contribution should be highlighted with 4-5 bulleted points for better and clear understanding.
5) Recent references need to be added. More references from 2022, 2023.

Experimental design

The experimental design procedure is not new. Several literature can be found in this domain. The authors required to find the design difference and advantages and dis advantages and should highlight their contribution. A complete block diagram of the system should be included to describe the complete process of the system for a better understanding.

Validity of the findings

The result analysis is done based on the given data base and 3 models.The authors are advised to to compare and validate on more dataset.

Additional comments

The writing, quality needs to be improved. The literature review should be re-written. More simulation result analysis should be provided to validate the model.

Cite this review as

---

## Round 0.2 · Minor Revisions

Thanks to the authors for their efforts to improve the work. However, it needs further discussions on the limitations of the research. Please continue to revise the article according to the comments. Then, it will be evaluated again.

Reviewer 1 ·

Basic reporting

No comment

Experimental design

No comment

Validity of the findings

Thank you for the opportunity to review this paper on demand prediction for urban air mobility (UAM). A few questions and comments for the authors:

What are the limitations of the machine learning method? What are the limitations of the NYC TLC data set? It would be helpful for the authors to discuss the applicability and/or limitations of surface taxi data being used for UAM air taxis. A few questions/potential limitations come to mind:
- Different price points (ground taxis are less expensive than anticipated price points for UAM)
- Ground taxis also can't fly point-to-point in the same way that an air taxi does - this can result in longer routing/travel times.

Based on the findings, are there any recommendations for policy and/or additional research?

Some additional resources that might be helpful for the authors:
https://www.mdpi.com/2071-1050/13/13/7421

Cite this review as

---

## Round 0.3 · accepted · Accept

Thanks to the authors for your efforts to improve the work. It successfully satisfied the reviewers. Congrats!

Reviewer 1 ·

Basic reporting

No comment

Experimental design

No comment

Validity of the findings

No comment

Additional comments

Authors have addressed reviewer comments.

Cite this review as